# Unveiling charge dynamics of visible light absorbing oxysulfide for efficient overall water splitting

Vikas Nandal [1,7 ✉], Ryota Shoji[2,7], Hiroyuki Matsuzaki [2 ✉], Akihiro Furube [3], Lihua Lin[4], Takashi Hisatomi [4], Masanori Kaneko [5], Koichi Yamashita [5], Kazunari Domen [4,6] & Kazuhiko Seki [1 ✉]

Oxysulfide semiconductor, $Y_2Ti_2O_5S_2$, has recently discovered its exciting potential for visible-light-induced overall water splitting, and therefore, imperatively requires the probing of unknown fundamental charge loss pathways to engineer the photoactivity enhancement. Herein, transient diffuse reflectance spectroscopy measurements are coupled with theoretical calculations to unveil the nanosecond to microsecond time range dynamics of the photogenerated charge carriers. In early nanosecond range, the pump-fluence-dependent decay dynamics of the absorption signal is originated from the bimolecular recombination of mobile charge carriers, in contrast, the power-law decay kinetics in late microsecond range is dominated by hole detrapping from exponential tail trap states of valence band. A well-calibrated theoretical model estimates various efficiency limiting material parameters like recombination rate constant, n-type doping density and tail-states parameters. Compared to metal oxides, longer effective carrier lifetime ~6 ns is demonstrated. Different design routes are proposed to realize efficiency beyond 10% for commercial solar-to-hydrogen production from oxysulfide photocatalysts.

[1] Global Zero Emission Research Center, National Institute of Advanced Industrial Science and Technology (AIST), Tsukuba 16-1 Onogawa, Tsukuba, Ibaraki 305-8569, Japan. [2] Research Institute for Material and Chemical Measurement, National Metrology Institute of Japan (NMIJ), National Institute of Advanced Industrial Science and Technology (AIST), Tsukuba 1-1-1 Higashi, Tsukuba, Ibaraki 305-8565, Japan. [3] Department of Optical Science, Tokushima University, 2-1 Minamijosanjima-cho, Tokushima 770-8506, Japan. [4] Research Initiative for Supra-Materials, Interdisciplinary Cluster for Cutting Edge Research, Shinshu University, 4-17-1 Wakasato, Nagano-shi, Nagano 380-8553, Japan. [5] Elements Strategy Initiative for Catalysts and Batteries (ESICB), Kyoto University, 1-30 Goryo-ohara, Nishikyo-ku, Kyoto 615-8245, Japan. [6] Office of University Professors, The University of Tokyo, 7-3-1 Hongo, Bunkyo-ku, Tokyo 113-8656, Japan. [7] These authors contributed equally: Vikas Nandal, Ryota Shoji. ✉email: nk.nandal@aist.go.jp; hiroyuki-matsuzaki@aist.go.jp; k-seki@aist.go.jp

Solar water splitting via particulate semiconducting photocatalysts is a promising renewable and sustainable technology to produce clean, green, and easily transportable hydrogen and oxygen fuels from large area panels[1–6]. For market commercialization, the most essential requirement for photocatalytic material is their ability to absorb visible solar spectrum, in addition to the efficient charge separation and extraction for high photocatalytic activity[7]. Inefficient charge separation and extraction with less external quantum efficiency (EQE) can be compensated by enhanced visible-light absorption or vice-versa to realize the desired solar-to-hydrogen (STH) efficiency (see Supplementary Fig. 1)[8]. Considering 100% EQE, the photocatalytic materials should be tailored for narrow band-gap energy <2.35 eV or light absorption edge >527 nm to achieve STH efficiency beyond 10% for practical applications.

Globally, various research groups engineered the band-gap energy for the development of visible-light absorbing metal-doped oxides-[9–11], (oxy)sulfides-[12–14], (oxy)nitrides-[15–18], (oxy) halides-[19–21], and metal chalcogenides-based photocatalysts for overall solar water splitting[22,23]. Recently, Wang et al. demonstrated the encouraging potentials of narrow band-gap $Y_2Ti_2O_5S_2$ photocatalyst, which absorbs large fraction of visible solar spectrum up to 650 nm, and generates a stochiometric durable $H_2$ and $O_2$ gas evolution[24]. However, the reported STH efficiency was quite lower (0.007%) than the theoretical limit (of 20.9%, Supplementary Fig. 2). Such low STH efficiency was attributed to low quantum yield due to high charge carrier recombination loss at the grain boundaries[24]. To date, the efficiency limiting recombination mechanisms and associated material parameters are unknown and not reported in the literature. Takata et al. realized the possibility of achieving an ideal EQE close to unity for a wide band-gap Al-doped $SrTiO_3$ photocatalyst by minimizing the recombination loss through the asymmetric or selective transport of charge carriers towards different crystal facets[25]. Generally, the charge carrier recombination competes with the charge transport and extraction, which dictates the device performance[26–32]. Therefore, it is imperative to probe the unexamined recombination processes for $Y_2Ti_2O_5S_2$ photocatalyst, and consequently, develop engineering strategies for enhanced efficiency. Compared with the competition between the recombination and transport processes, less attention has been paid for the characterization of band-tail states and their impact on the quantum efficiency for photocatalytic materials.

Herein, transient diffuse reflectance spectroscopy (TDRS) is coupled with theoretical calculations to reveal the origin of distinct characteristics in the early nanosecond and late microsecond time range, respectively. With a calibrated model, the unreported material parameters are determined to facilitate design principles towards the realization of highly efficient and stable performing $Y_2Ti_2O_5S_2$ photocatalyst.

## Results and discussion

**Sample characterization.** $Y_2Ti_2O_5S_2$ photocatalyst was synthesized by solid-state-reaction fabrication process (Methods). Scanning electron microscopy (SEM) image of the sample in Fig. 1a indicates different $Y_2Ti_2O_5S_2$ particle sizes ranging from few to tens of micrometer. From Fig. 1b and Supplementary Fig. 3, X-ray diffraction (XRD) pattern suggests that the particles exhibit single and pure crystalline phase with tetragonal crystal symmetry (with space group I4/mmm)[33]. The measured XRD pattern is well in accordance with the simulated data of $Y_2Ti_2O_5S_2$. Diffuse reflectance spectroscopy (DRS) measurement, in Fig. 1c, shows visible-light absorption up to wavelength of 650 nm due to relatively narrow band gap of nearly 1.9 eV. Similar characteristics of XRD pattern and DRS were recently

reported by our group[24]. From density functional theory (DFT) calculations, in Supplementary Fig. 4a, the conduction band (CB) minimum and valence band (VB) maximum are primarily contributed from Ti-3d and S-3p orbitals, respectively. Band structure calculation in Supplementary Fig. 4b reveals that $Y_2Ti_2O_5S_2$ is a direct band-gap semiconductor, as pointed out previously[34]. The partial density of states computed using HSE06 hybrid functional (Supplementary Fig. 5) displays an energy band gap of 1.91 eV, which is well in agreement with the reported values of 1.9 eV (measured)[24] and 2.19 eV (DFT/HSE06 method)[34]. The slight discrepancy in the values of the band gap might be attributed to the difference in the basis sets. The calculated optical properties such as dielectric function, complex refractive index, reflectivity, and absorption coefficient of $Y_2Ti_2O_5S_2$ are provided in Supplementary Fig. 6. The simulated reflectivity spectra coincide with the measured spectrum, which confirms the validity of the computed optical properties. From Fig. 1d or Supplementary Fig. 6f, the absorption coefficient $\alpha_{abs}$ for $Y_2Ti_2O_5S_2$ ranges from $10^3 \, cm^{-1}$ to $10^5 \, cm^{-1}$ within the ultraviolet–visible solar spectrum from 2.4 to 4 eV. In addition, Supplementary Fig. 7 displays that the $\alpha_{abs}$ of $Y_2Ti_2O_5S_2$ is similar to that of other well-studied visible-light absorbing oxysulfide ($La_5Ti_2CuS_5O_7$)[35], however, one order less than the metal oxides ($BiVO_4$; $\alpha$-$Fe_2O_3$)[36,37] and nitride ($Ta_3N_5$)[27,28] photocatalysts. Consequently, the oxysulfides absorb the ultraviolet solar spectrum from 3.1 to 4.2 eV within the depth of 50 nm to 600 nm, whereas metal oxides and nitride absorb within 10 to 50 nm from the electrolyte interface.

**Transient diffuse reflectance spectroscopy.** Transient diffuse reflectance spectroscopy (TDRS; Methods) measurements were performed to unravel the physical origin behind charge carrier dynamics of the $Y_2Ti_2O_5S_2$ photocatalyst. TDRS is essentially similar to transient absorption spectroscopy. In the TDRS measurements, transient absorption signal was detected in diffuse reflection mode because of the opaque nature of the photocatalyst powder samples. Schematic illustration and related discussion for TDRS measurements are provided in Supplementary Fig. 8. Figure 1e displays that the rate of decay of absorption signal is almost similar across entire probe photon energy spectrum. These features suggest band-to-band relaxation of charge carriers without irreversible trapping into deep trap levels within the time scale of measurement. As electrons and holes are generated and recombined in pairs, the kinetics of both species should be the same. Within uncertainty in assignment of transient diffuse reflectance (TDR) spectra for shallowly trapped carrier components, the spectra could be assumed as a linear combination of that originating from the total concentration of holes and electrons; both charge species show similar kinetics. Therefore, we analyzed the probed decay kinetics using the total concentration of electrons. TDRS measurements were performed to obtain absorption characteristics over wide range of delay time $t$ (i.e., from ps to μs) at a pump and probe photon energy of 3.1 and 0.24 eV, respectively. The intensity of pump fluence $P_{FL}$ was varied from 0.075 to 4.5 μJ per pulse. In Fig. 1f, the absorption characteristics exhibit distinct $P_{FL}$-dependent fast decay in early time range (ns) in comparison to relatively slow power-law decay in late time range (μs). Such contrasting decay characteristics indicate the possibility of different relaxation processes in respective time scales. As demonstrated in several previous studies[36,38–40], in transient absorption and reflectivity measurements, transient heating of the sample induced by the pump laser pulse can cause thermal components in the measured spectra and a proper assessment and isolation of its components are essential for accurate interpretation of photoinduced (nonthermal) electronic responses. To this end, we estimated the effect of pump-

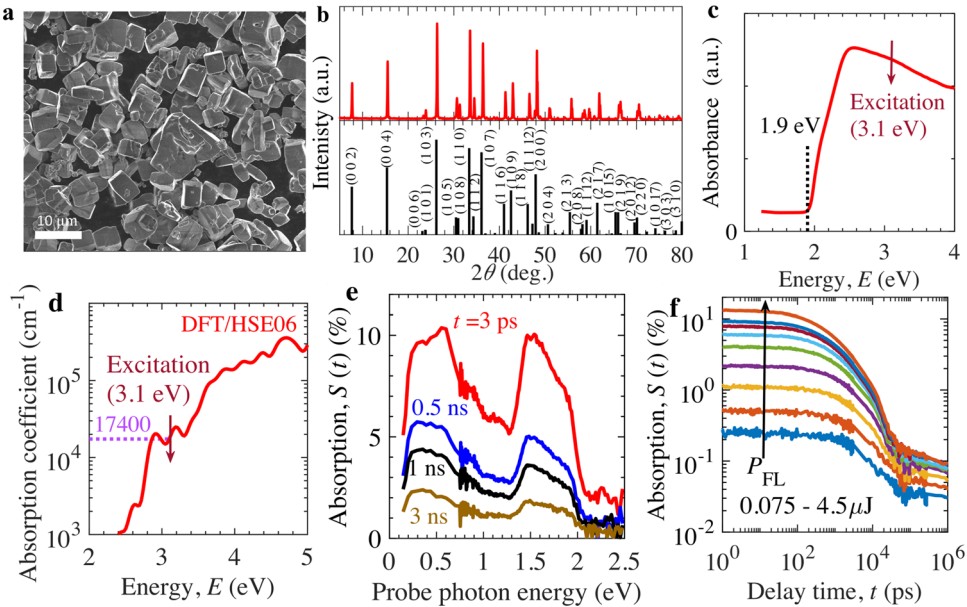

**Fig. 1 Morphology, structure, and absorption spectroscopy of Y$_2$Ti$_2$O$_5$S$_2$ photocatalyst. a** SEM image. **b** XRD pattern (red: measured, black: simulated). **c** Diffuse reflectance spectrum with estimated band-gap energy of 1.9 eV. **d** Absorption coefficient $\alpha_{abs}$ from DFT/HSE06 calculations. **e** TDR spectra at pump fluence intensity $P_{FL} = 3\,\mu J$. **f** TDR kinetics at probe photon energy of 0.24 eV. Source data are provided as a Source Data file.

induced heating on TDR signal and concluded that the TDR signal of Y$_2$Ti$_2$O$_5$S$_2$ probed at 0.24 eV reflects only photoinduced electronic processes as shown in the Supplemental Note 1 (Supplementary Figs. 9 and 10).

**Early-time kinetics.** Figure 2 presents the analysis of TDR signal, $S(t)$, decay characteristics in early ns time range. Figure 2a shows that the decay rate increases with the increase in pump fluence, which is the usual signature of the bimolecular charge carrier recombination[41]. The bimolecular recombination rate can be expressed as $d\Delta n/dt = -k_r(n_{eq} + \Delta n)\Delta n$, where, $k_r$, $n_{eq}$, and $\Delta n$ are the bimolecular recombination rate constant, n-type doping or equilibrium electron density, and excess charge carrier density, respectively. If trap levels are present and detrapping can be ignored up to few nanoseconds, the early-time kinetics is more precisely given by $d\Delta n/dt = -k_r(n_{eq} + N_t + \Delta n)\Delta n$, where $N_t$ indicates the density of states of the shallow traps. Considering $S(t) = \beta\Delta n$, the rate relation can be rearranged as $dS(t)/dt = -k_r(n_{eq} + N_t + S(t)/\beta)S(t)$, which is solved to obtain $S(t)$ as[41]

$$\frac{1}{S(t)} = \frac{1}{S(0)} + \left(\frac{k_r}{\beta} + \frac{k_r(n_{eq} + N_t)}{S(0)}\right)t. \quad (1)$$

The first term on the right-hand side of above equation is termed as intercept $1/S(0)$, whereas the second term is the product of slope (which varies linearly to the intercept $1/S(0)$) and time $t$. In addition, the slope depends on proportionality constant $\beta$, $k_r$, $n_{eq}$, and $N_t$. Figure 2b shows that the maximum TA signal $S(0)$ increases almost linearly with $P_{FL}$ or incident photon density $I_p$. With absorption coefficient of $17,400\,cm^{-1}$ (Fig. 1d), the initially photogenerated charge carrier density $\Delta n_0 = 7.90 \times 10^{18}\,cm^{-3}$ at $P_{FL}$ of 3 µJ is estimated. In Fig. 2b, $S(0) = \beta\Delta n_0$ in low $P_{FL}$ range (0–2.4 µJ) provides $\beta = 1.30 \times 10^{-18}\,cm^3$. In Fig. 2c, inverse absorption transient shows linearly increasing

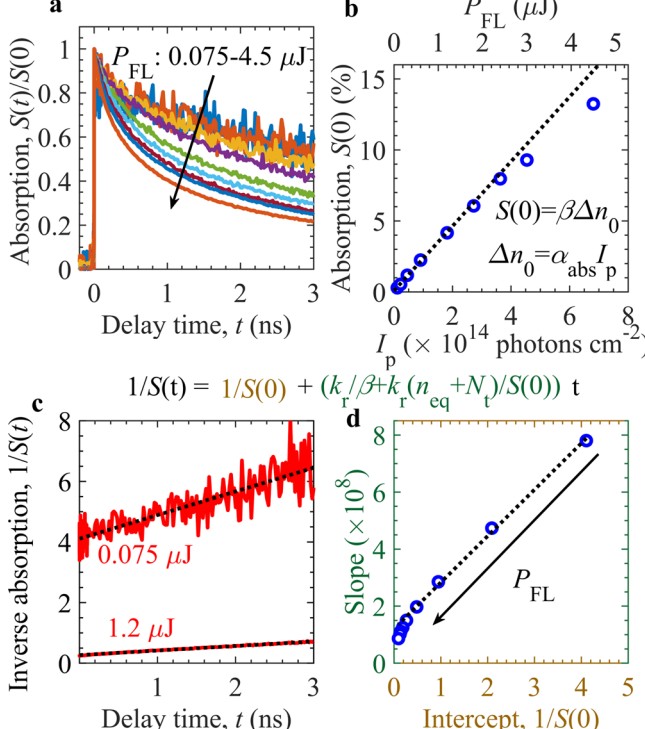

**Fig. 2 Early-time decay analysis of TDR signal of Y$_2$Ti$_2$O$_5$S$_2$ photocatalyst. a** Normalized TDR signal $S(t)$ decay dynamics at probe photon energy of 0.24 eV. **b** Maximum TDR signal $S(0)$ against pump fluence intensity $P_{FL}$ and incident photon density $I_p$. **c** Inverse TDR profile fitted with the analytical model of $1/S(t)$, shown on top of (**c**, **d**), to obtain slope and intercept plot in **d**. In **b–d**, the black dotted lines correspond to respective linear fits. The arrow direction in (**a**, **d**) represents increase in $P_{FL}$. Source data are provided as a Source Data file.

behavior with $t$ at different $P_{FL}$, as per Eq. (1). The results indicate that the defined slope and intercept decrease linearly with the increase in $P_{FL}$. The estimated slope and intercept, from the linear fit of Eq. (1) to the experimental data at various $P_{FL}$ in Fig. 2c, is plotted on $y$- and $x$-axis of Fig. 2d, respectively. Linear fit in low $P_{FL}$ regime in Fig. 2d is further employed to determine $k_r(n_{eq} + N_t)$ and $k_r/\beta$ from slope and intercept, respectively. From early-time TA decay analysis in Fig. 2, the calculated values of $k_r$ and $n_{eq} + N_t$ are $1.57 \times 10^{-10}$ cm$^3$s$^{-1}$ and $1.03 \times 10^{18}$cm$^{-3}$, respectively. In the following, we show that $n_{eq}$ and $N_t$ can be disentangled from analysis of late-time decay kinetics.

**Late-time kinetics.** Late-time TDR decay in Fig. 3a demonstrates that the TDR signal $S(t)$ follows power-law decay with time, which is governed by relation $S(t) = A \times t^{-\alpha}$. In Fig. 3b, c, the estimated amplitude $A$ reduces with the decrease in $P_{FL}$, whereas the exponent $\alpha$ is almost independent of $P_{FL}$ (except at low $P_{FL}$) and is clearly smaller than 1 expected from bimolecular recombination (as $S(t) \propto t^{-1}$ in Eq. (1)). The reduction of $\alpha$ from 1 could be the effect of shallow trap states. The shallow trap states near VB are considered to participate in trapping and detrapping of holes, however, electrons filled trap states near CB do not contribute towards trapping/detrapping of electrons due to strong

n-type characteristics of Y$_2$Ti$_2$O$_5$S$_2$. The Mott-Schottky (MS) analysis is provided in Supplementary Fig. 11 to confirm that Y$_2$Ti$_2$O$_5$S$_2$ is a heavily n-type doped photocatalyst. For this, the capacitance of Y$_2$Ti$_2$O$_5$S$_2$ electrode (prepared by particle-transfer method) was normalized to the projected Ti substrate area, which was less than the total surface area of the Y$_2$Ti$_2$O$_5$S$_2$/electrolyte interface[24]. Considering this, the extracted doping density from MS analysis provides an upper limit of n-type doping density $N_d^{max}$. The results indicate that $N_d^{max}$ is $1.9 \times 10^{20}$ cm$^{-3}$ and consequently, the Fermi-energy level $E_f$ is expected to be close to the CB energy minimum of $E_c$, as per Fermi-Dirac statistics. In literature, multiple trapping models have been developed to explain that the power-law decay is originated from the energy dispersive tail states near CB and/or VB for intrinsic semiconductors[42–47]. Similar power-law decay with trap-limited recombination has been observed for other n-type semiconductors[48–50]. The $\alpha$ was often termed as the dispersive parameter and was employed to quantify characteristic energy $E_0$ of exponential tail states (using $E_0 = k_B T/\alpha$, $k_B$ is the Boltzmann constant and $T$ is the temperature). As a result, we formulate the multiple trapping model for holes in heavily n-type doped Y$_2$Ti$_2$O$_5$S$_2$ semiconductor. Figure 3d displays the detailed model description of governing charge relaxations in Y$_2$Ti$_2$O$_5$S$_2$. Here, we consider bimolecular recombination of mobile charge carriers along with trapping and detrapping of holes as the dominant mechanisms behind decay dynamics. Trap-assisted recombination is insignificant as evident from Fig. 2, and therefore, not considered in the proposed model. Exponential tail-states of VB, with trap density $N_t$ and $E_0$, are introduced, which lead to the trapping and detrapping of holes. As shown in the Supplementary Notes 2 and 3, the trapped hole density $p_t^A(t)$ is obtained as

$$p_t^A(t) \approx \frac{N_t \pi \alpha}{[1 + (n_{eq}/\Delta n_0)]\Gamma(1 - \alpha)\sin(\pi\alpha)(k_t N_m t)^\alpha},  \quad (2)$$

where, $\Gamma(x)$ is the gamma function[51]; $N_m$ is the effective density of states for VB, and $k_t$ is the trapping rate constant for mobile holes. The trapped holes $p_t^A(t)$ is calibrated to the experimental data in late-time decay kinetics, which provides the initial estimates of $\alpha = 0.19$, $E_0 = 0.137$ eV, and $N_t = 5.1 \times 10^{17}$cm$^{-3}$.

**Theoretical model calibration.** Figure 4 shows the model calibration with the TDR signal of Y$_2$Ti$_2$O$_5$S$_2$ photocatalyst. For simplicity, the initial condition at time $t = 0$ for the trapped hole

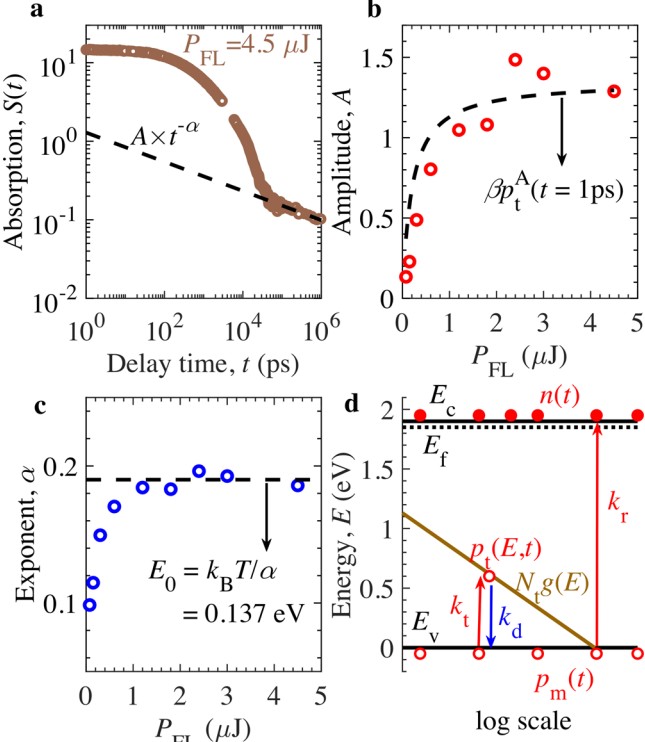

**Fig. 3 Late-time decay analysis of TDR signal of Y$_2$Ti$_2$O$_5$S$_2$ photocatalyst. a** TDR profile at pump fluence intensity $P_{FL} = 4.5$ μJ and probe photon energy of 0.24 eV fitted with the power law: $A \times t^{-\alpha}$. **b, c** Variation of estimated amplitude $A$ and exponent $\alpha$ with $P_{FL}$. In **b** and **c**, the dashed lines correspond to the analytical solution from Eq. (2) at trap density $N_t = 5.1 \times 10^{17}$ cm$^{-3}$ and energy dispersion parameter $\alpha = 0.19$ ($E_0 = 0.137$ eV). **d**, Proposed model schematic showing exponential tail-states of VB ($N_t g(E)$) along with bimolecular band-to-band transition of mobile electrons and trapping/detrapping of holes. Here, $k_r$, $k_t$, and $k_d$ are the corresponding rate constants, whereas $p_m(t)$, $p_t(E, t)$, and $n(t)$ are the density of mobile holes, trapped holes at energy $E$, and electrons at time $t$ after photoexcitation, respectively. Source data are provided as a Source Data file.

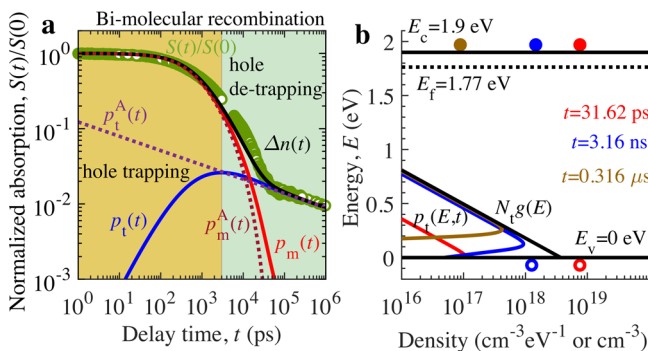

**Fig. 4 Theoretical analysis of decay kinetics of TDR signal of Y$_2$Ti$_2$O$_5$S$_2$ photocatalyst. a** Numerical (solid line) and analytical model (dotted line) calibration with the measured normalized TDR signal $S(t)/S(0)$ (circle) measured at pump fluence intensity $P_{FL} = 3$ μJ and probe photon energy of 0.24 eV. **b** Mapping of calculated trapped hole density $p_t(E, t)$ in cm$^{-3}$ eV$^{-1}$ across energy $E$, mobile electron (solid circles) and hole density (open circles) in cm$^{-3}$ at different decay time $t$. Source data are provided as a Source Data file.

density $p_t(E, 0)$ at energy $E$ (from VB edge) and hence integrated trapped hole density $p_t(0)$ is set to 0, whereas the mobile hole density $p_m(0)$ or $p_i = \Delta n_0$ is estimated using Lambert–Beer law. The late-time asymptotic decay is insensitive to the choice of $p_t(0)$ as shown in Supplementary Fig. 12. Material parameters employed for theoretical calculations are provided in Supplementary Table 1. These calculations enable us to probe the charge carrier (i.e., mobile electrons, mobile holes, and trapped holes) kinetics behind TDR decay dynamics. The results in Fig. 4a demonstrate that the TDR signal is in excellent agreement with the theoretical solutions of the model. In addition, in Supplementary Fig. 12, the numerical solutions match well with the experimental TDR signals for various $P_{FL}$ to further validate our proposed model. As predicted earlier, the early-time decay kinetics of TDR (up to 3 ns) is influenced by mobile electron and hole decay, which is resulted from the bimolecular recombination. However, the late-time decay kinetics around sub-microsecond time range is dominated by relatively slower electron decay dynamics, which is governed by detrapping of holes from VB tail states. Figure 4b presents the mapping of numerically calculated time evolution of charge carrier density such as mobile electron $n(t)$, hole $p_m(t)$, and trapped hole density $p_t(E, t)$. In ns time range, the trapping of mobile holes results in the uniform increase of $p_t(E, t)$ across energy $E$ and hence $p_t(t)$. In sub μs-time range, the detrapping of holes proceeds to the significant decrease of $p_t(E, t)$ from shallow VB tail states (0–0.3 eV). When $p_t(t)$ decays in sub μs-time range in Fig. 4a, the hole detrapping does not affect the population of trapped hole density in relatively deeper tail states (>0.3 eV) due to insufficient thermal energy $k_B T$ (as detrapping rate constant $k_d = k_t \exp(-E/k_B T)$). If the mobile hole decays solely by bimolecular recombination with electrons, the density of holes decays by power law with the exponent of −1 at late times. The asymptotic power-law decay is delayed owing to trapping and detrapping of holes from the VB tail states. When the trap density is slowly varying with $E$ in exponential tail states of VB, the delay influences the exponent of the power-law decay. Further, we believe that the dispersion parameter $\alpha$ primarily depends on the $E_0$ of shallow tail states around peak of $p_t(E, t)$ in Fig. 4b, where the power-law decay of hole density is observed.

**Design routes for performance optimization.** The bimolecular recombination rate constant of $1.57 \times 10^{-10}\,cm^3 s^{-1}$ for $Y_2Ti_2O_5S_2$ is comparable with the well-established direct band-gap semiconductors like GaAs ($\sim 0.5 - 1 \times 10^{-10}\,cm^3 s^{-1}$)[52,53] and $CH_3NH_3PbI_3$ perovskite ($\sim 1 - 10 \times 10^{-10}\,cm^3 s^{-1}$)[54,55]. Based on the bimolecular recombination rate constant and the other physical parameters, we show that $Y_2Ti_2O_5S_2$ has potential prospects as photocatalysts for overall solar water splitting by performance optimization. In principle, the photogenerated electrons and holes within a diffusion length from the reaction site participate in water splitting to produce hydrogen and oxygen, respectively. The charge carrier diffusion length can be defined by $L_D = \sqrt{D\tau_{eff}}$ with diffusion constant $D = \mu k_B T/q$ (as per Einstein relation), where $\mu$ and $q$ are the charge carrier mobility and electronic charge, respectively[56]. Since TDR analysis reveals that the $Y_2Ti_2O_5S_2$ photocatalyst exhibits strong signature of bimolecular recombination and trapping of photogenerated charge carriers with no evidence of Shockley-Read-Hall and auger recombination, the effective bulk carrier lifetime $\tau_{eff}$ of $Y_2Ti_2O_5S_2$ is given by $\tau_{eff} = 1/(k_r n_{eq} + k_t N_t)$ as $n_{eq} \gg \Delta n_0$ under AM 1.5G operating condition. Using the values obtained from the TDR analysis, we find $\tau_{eff} = 6.14$ ns, which is higher than well-known visible-light absorbing metal oxides or nitrides like $BiVO_4$[50,57],

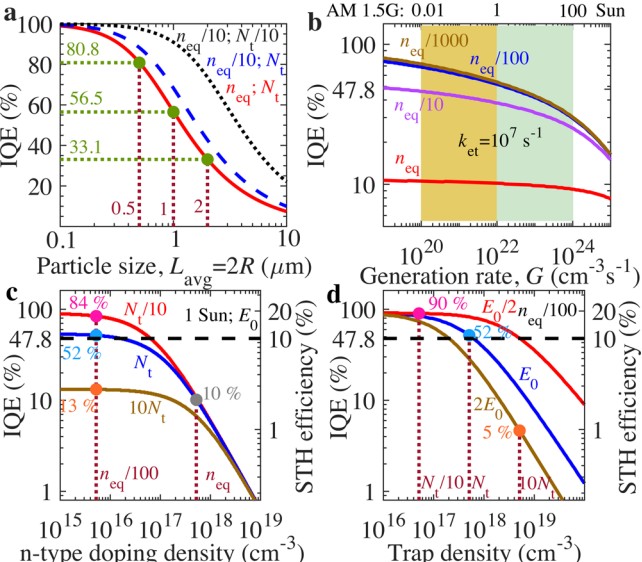

**Fig. 5 Optimization design strategies to realize high efficiency for $Y_2Ti_2O_5S_2$ photocatalyst. a** Impact of particle size, n-type donor density $n_{eq}$ and trap density $N_t$ on internal quantum efficiency (IQE). **b** Influence of $n_{eq}$ at various charge carrier generation rate $G$ for charge extraction rate $k_{et} = 10^7\,s^{-1}$. **c, d** Effect of $n_{eq}$, $N_t$, and characteristic energy $E_0$ on the IQE and STH efficiency of $Y_2Ti_2O_5S_2$ photocatalyst. Here, $n_{eq}$, $N_t$, and $E_0$ correspond to the estimated values (in Supplementary Table 1) from the TA decay kinetics. The IQE and STH efficiency is calculated from Eq. (3) and Supplementary Note 5. Source data are provided as a Source Data file.

$Cu_2O$[58], $\alpha$-$Fe_2O_3$[59], and $Ta_3N_5$[41]. Considering typical value of the effective charge carrier mobility $\mu = 1\,cm^2 V^{-1} s^{-1}$, the diffusion length $L_D \approx 126$ nm, which is significantly smaller than the synthesized average radius ($R$) of almost 5 μm of $Y_2Ti_2O_5S_2$ particles. This suggests that a significant fraction of photogenerated charge carrier recombines and is trapped in tail states of VB. By assuming uniform charge carrier generation inside spherical particles of $Y_2Ti_2O_5S_2$ with uniform coverage of reactive sites on the surface of particles, the fraction of the photogenerated mobile carriers transported to the surface for respective gas evolution reaction can be estimated from $4\pi R^2 L_D/(4\pi R^3/3)$ for $L_D \ll R$; by substituting $L_D \approx 126$ nm and $R \approx 5$ μm, we find only 7% (= $100 \times 3L_D/R$), which can be considered one of the prime reasons of realized low apparent QE. As shown in the Supplementary Note 4, the expression of the internal quantum efficiency (IQE) can be generalized as

$$IQE = (3L_D/R)\left[\coth(L_D/R) - L_D/R\right]. \quad (3)$$

In the limit of $L_D/R \ll 1$, $3L_D/R$ is recovered. Both $n_{eq}$ and $N_t$ should be reduced to increase charge carrier diffusion length $L_D$. Figure 5 presents the performance optimization design strategies to achieve desired IQE and STH efficiency beyond 47.8% and 10%, respectively, for commercial application of $Y_2Ti_2O_5S_2$ photocatalyst. Based on Eq. (3), Fig. 5a displays that the IQE increases substantially from 33.1% (or 7%) to 80.8% as the particle size reduces from 2 μm (or 10 μm) to 500 nm, however, additional IQE improvement requires the increase in the diffusion length $L_D$ by 10-fold reduction of $n_{eq}$ and/or $N_t$. For particle size less than the charge diffusion length, the IQE could be estimated by considering charge extraction rate at the photocatalyst surface ($k_{et}$) and the recombination rate $k_r n_{eq}$ as $k_{et}/(k_{et} + k_r n_{eq})$ as shown in the Supplementary Note 5. With the estimates (from TDRS), in Fig. 5b and Supplementary Fig. 13, the IQE is almost independent of generation rate $G$. However, the

IQE reduces substantially with the increase in $G$ for low $n_{eq}$ as per derived Supplementary Eq. (S50) due to the VB tail states. Stable IQE for overall water splitting by reducing VB tail-states parameters (i.e., $E_0$ and $N_t$) is imperative for durable and robust operation of photocatalyst at different weather conditions and geographical locations, in addition to solar concentrator applications.

Under AM 1.5G illumination, the results in Fig. 5c suggest that the IQE and STH efficiency improve significantly and almost gets saturated with the decrease in $n_{eq}$ by reducing bimolecular charge recombination loss. However, the maximum improvement in IQE with $n_{eq}$ reduction depends on the $N_t$. For instance, for extracted $N_t = 5.1 \times 10^{17} \mathrm{cm}^{-3}$, the IQE improves tremendously from 10% to 52% with the 100-fold reduction in $n_{eq}$. Further improvement (or reduction) in IQE from 52% to 84% (13%) is predicted with the 10-fold decrease (increase) of $N_t$ for characteristic energy $E_0 = 0.137$ eV. Such trap-density-induced IQE decrease can be suppressed by decreasing the characteristic energy for deep tail states $2E_0$ to shallow tail states $E_0/2$ as evident in Fig. 5d. The desired STH efficiency of 10% for commercialization can potentially be realized with the improvement in IQE to 47.8% by designing $Y_2Ti_2O_5S_2$ photocatalyst to achieve at least 10-fold reduction in $n_{eq}$ along with $N_t$. However, further improvement in STH efficiency beyond 15% requires 100-fold reduction of $n_{eq}$ and 10-fold decrease in $N_t$.

In summary, transient diffuse reflectance spectroscopy (TDRS) measurements were coupled with theoretical calculations to unravel physical insights behind the photogenerated charge carrier dynamics of particulate $Y_2Ti_2O_5S_2$ photocatalyst. We demonstrated distinct pump-fluence-intensity-dependent carrier decay kinetics of the transient absorption signal ($\propto t^{-1}$) in early-time range and ($\propto t^{-0.19}$) in late-time range. In particular, the early-time decay dynamics in nanosecond range was attributed to the bimolecular charge carrier recombination. However, the power-law decay kinetics in microsecond range was limited by hole detrapping from the exponential tail states of valence band. With theoretical model calibration, the performance influencing material parameters were determined. In addition to other physical parameters, TDRS analysis enables the estimate for n-type doping density of $Y_2Ti_2O_5S_2$ particulate system, which cannot be accurately measured by conventional means (applicable to thin-film electrodes) of Hall measurements, Mott-Schottky analysis, and spectroscopic ellipsometry. We introduced optimization design strategies of controlling particle size, n-type doping density, tail-trap states to realize high and stable internal quantum efficiency for overall water splitting of $Y_2Ti_2O_5S_2$. Our work provides a fundamental benchmark towards the understanding of dominant charge loss mechanisms for the development of efficient oxysulfide photocatalysts with the potentials of achieving STH efficiency beyond 10% for commercial photocatalytic overall water splitting.

## Methods

**Sample preparation**. The $Y_2Ti_2O_5S_2$ powder was prepared by a solid-state reaction. $Y_2O_3$ (Wako Pure Chemical Industries, 99.99%), $Y_2S_3$ (High Purity Chemicals, 99.9%), and $TiO_2$ (Rare Metallic, 99.99%) were mixed at a ratio of 1:2:6 in an argon-filled recirculating glovebox with an $O_2$ concentration of less than 3 ppm. The dew point was lower than 193 K. To obtain a sulfur-rich environment during the reaction, sulfur powder (High Purity Chemicals, 99.99%, 5 wt% with respect to the total of the other starting materials) was added to the precursor. The resulting mixture was sealed in an evacuated quartz tube and calcined in a muffle furnace. The calcination temperature was increased from room temperature to 773 K at a rate of 5 K min$^{-1}$, elevated to 873–1,073 K at 1 K min$^{-1}$, and maintained at the target value for 96 h before natural cooling. To remove the sulfur species adsorbed on the surface, the $Y_2Ti_2O_5S_2$ powder was annealed in air at 473 K for 1 h, thoroughly rinsed with distilled water and dried in vacuum at 313 K.

**Scanning electron microscopy**. The morphology of the sample was studied using JEOL JSM-7600F field emission scanning electron microscope (SEM).

**X-ray diffraction**. X-ray diffraction (XRD) measurement was performed on Rigaku MiniFlex 300 with Cu Kα1 radiation ($\lambda = 1.5406$ Å).

**Diffuse reflectance spectroscopy**. Diffuse reflectance spectroscopy (DRS) measurements were made using an ultraviolet–visible–near–infrared spectrometer (V-670, JASCO) and were converted from reflectance into the Kubelka–Munk (KM) function.

**Normal reflectance spectroscopy**. Normal reflectance spectroscopy measurements were performed on the crystal face of particle with size of 10–20 μm using an ultraviolet–visible–near–infrared spectrometer with an optical microscope (MSV-5200, JASCO).

**Transient diffuse reflectance spectroscopy**. Transient diffuse reflectance spectroscopy (TDRS) is essentially similar to transient absorption (TA) spectroscopy. In the TDR measurements, TA signal was detected in diffuse reflection mode because of the opaque nature of the photocatalyst powder samples. The TA intensity in diffuse reflection mode is presented herein in units of percentage absorption (Absorption (%)), calculated as $100 \times (1 - R/R_0)$, where $R$ and $R_0$ are the intensities of the diffusely reflected light with and without pump excitation, respectively.

Femtosecond TDRS measurement ($t < 3$ ns) were carried out using a Ti: sapphire laser with a regenerative amplifier (Spectra-Physics, Solstice, wavelength of 800 nm, pulse width of 100 fs, pulse energy of 3.5 mJ per pulse and repetition rate of 1 kHz) as a light source. The output from the laser was split into four paths for the excitation of two optical parametric amplifiers (OPAs: Spectra-Physics, TOPAS Prime), the white-light-continuum generation by focusing the fundamental light (800 nm) into a sapphire plate, and the second- and third-harmonic generations of the fundamental light (800 nm) by using BBO (β-BaB$_2$O$_4$) crystals. The second-harmonic light (400 nm) was used for pump light, and pump light intensity was varied by neutral density filters from 0.075 μJ/pulse to 4.5 μJ/pulse. For the probe light ranging from 440 nm to 8530 nm, a white-light-continuum covering from 440 nm to 1600 nm and infrared (IR) light longer than 1600 nm generated from OPA equipped with difference-frequency generation crystal was used. The delay time of the probe pulse relative to the pump pulse was controlled up to 3 ns by changing the optical path length of the pump pulse. The time resolution of the system was about 140 fs. Powder YTOS samples were taken in CaF$_2$ cuvette whose size is $45 \times 10 \times 1.0$ mm. A Si amplified photodetector (Thorlabs, PDA36A-EC) and an InGaAs photodetector (Thorlabs, PDA20CS-EC) was used to probe for 440~1100 nm and 1100~1600 nm, respectively. For the probe from 440 nm to 1600 nm, the diffusely reflected light from the sample was passed through a grating monochromator (Princeton Instruments, Acton SP2150) for data acquisition. A liquid-nitrogen-cooled HgCdTe photodetector (Kolmar Technologies, KMPV11-1-J1) was used to probe for 1600–8530 nm. The diameter of the pump beam on the sample was about 1 mm and the irradiated area of the pump beam was estimated using a beam profiler (Newport LBP2-HR-VIS2).

In TDRS measurements for $t > 3$ ns, the continuous-wave IR light (5250 nm) from a quantum cascade laser (Thorlabs, QD5250CM1) was used as the probe light source. The pump light of 400 nm was identical to that used in the measurements for $t < 3$ ns described above. The diffusely reflected light from the sample was detected by a liquid-nitrogen-cooled fast HgCdTe photodetector (Kolmar Technologies, KV104-0.25-A-2/11, bandwidth of 80 MHz). The signal from the detector was amplified with a voltage amplifier (Femto, DHPVA-200) and subsequently processed and recorded with a digital oscilloscope (Lecroy, WaveRunner 6200 A). The pump-induced signal (AC signal) was selectively extracted by using the AC-coupled mode of the amplifier. The DC offset of the signal from the detector, on the other hand, was independently recorded with a digital multimeter (National Instruments, USB-4065) in order to calculate the Absorption (%). As a result, very small TA signals (<0.01%) were detected with few ns time resolution.

**Density functional theory (DFT) calculations**. Using open-source Quantum espresso tool[60], ab initio first-principle calculations of density functional theory (DFT) of $Y_2Ti_2O_5S_2$ were performed to calculate of partial density of states (PDOS) along with the band structure (BS). For this, tetragonal crystal symmetry of $Y_2Ti_2O_5S_2$ in Supplementary Fig. 4 with space group I4/mmm was considered as a model where the unit cell contains 4 atoms of Y, Ti, S each and 10 atoms of O. Ultrasoft pseudopotentials along with Perdew–Burke–Emzerhof (PBE) exchange correlation functional having kinetic energy cutoff 25 Ry and energy for localized charge density cutoff 225 Ry of respective elements were employed for geometry relaxation and DFT calculations. Broyden, Fletcher, Goldfarb, and Shanno (BFGS) algorithm based on quasi newton iterative solver was used for ion and cell relaxation. The threshold for force and electron kinetic energy was set to $10^{-4}$ Ry/Bohr and $10^{-7}$ Ry, respectively and the optimized lattice parameters $a = b = 3.74$ Å, and $c = 22.59$ Å were obtained at minimum total energy of $-1220.678$ Ry. Using optimized cell geometry, with k-mesh points (7 7 7), self-consistent followed by non-self-consistent field simulations were done to obtain PDOS and BS in Supplementary Fig. 4a, b, respectively.

The structural optimizations, the analysis of dielectric function of $Y_2Ti_2O_5S_2$ were performed using the VASP package[61]. The projected augmented wave (PAW)[62]

pseudopotential with exchange and correlation functional under generalized gradient approximation in the Perdew–Burke–Ernzerhof (PBE)[63] form was used in the calculations. A plane-wave basis set with the kinetic energy cutoff of 520 eV was used for the expansion of the electronic wave functions. The Brillouin zones were sampled in the G centered k-point grids of $8 \times 8 \times 1$. The electronic and force convergence criteria during the structural optimization were set to $10^{-8} \times$ eV and $10^{-2} \times$ eVÅ$^{-1}$, respectively and the optimized lattice parameters $a = b = 3.79$ Å and c $= 23.0$ Å were obtained. The Heyd–Scuseria–Ernzerhof functional (HSE06)[64] with the Hartree-Fock screening parameter of 0.2 was used to compute the partial density of states along with the dielectric function in the independent-particle picture in Supplementary Figs. 5 and 6.

## Data availability
The source and raw data generated in this study are provided in the Source Data file. Source data are provided with this paper.

## Code availability
The code that supports the findings of this work are available from the corresponding authors upon reasonable request.

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

## Acknowledgements

This work was supported by the "Research Project for Future Development: Artificial Photosynthetic Chemical Process (ARPChem)" (METI/NEDO, Japan: 2012–2022). M. K. and K. Y. acknowledge the support by JSPS KAKENHI in Scientific Research on Innovative Areas "Innovations for Light-Energy Conversion (I$^4$LEC)". We acknowledge Dr. Tatsuya Miyamoto and Mr. Naoki Takamura in The University of Tokyo for help with normal reflectance spectroscopy measurements.

## Author contributions

H.M., T.H., K.S. and K.D. conceived the idea and initiated the research. L.L. with the help of T.H. and K.D. prepared the sample and characterized materials by SEM, XRD, and DRS measurements. M.K., V.N. and K.Y. performed DFT calculations. R.S. and H.M. performed TAS experiments. V.N. and K.S. performed the theoretical modeling and analyzed the data with R.S., H.M. and A.F. V.N. and K.S. wrote the manuscript with contributions from the other authors. All authors contributed to the scientific discussion and editing of the manuscript.

## Competing interests

The authors declare no competing interests.
