## [Peer Review File · Nature Communications]

REVIEWER COMMENTS

Reviewer #1 (Remarks to the Author):

This manuscript presents a transient absorption study of Y₂Ti₂O₅S₂, an oxysulfide semiconductor that was recently shown to drive overall water splitting under visible-light irradiation. The authors present some sophisticated modeling of the transient absorption data to describe both the early-time kinetics (attributed to bimolecular recombination of mobile charge carriers) and late-time kinetics, which exhibit power law behavior and are attributed to detrapping of holes from band-tail states. The authors also perform a quantitative analysis of how various material parameters are expected to impact the performance of Y₂Ti₂O₅S₂ in photoinduced hydrogen generation.

Overall, despite the complexity of their model and its ability to fit the TA data, the only insight provided that may explain the improved photocatalytic activity of this specific oxysulfide material compared to other photocatalysts, such as metal oxides, is a longer effective lifetime of the initial photoexcited state of ~6 ns. However, this value and the values obtained for various carrier and trap state densities is based on the assumption that the absorption coefficient is $\sim 10^5 \text{ cm}^{-1}$. Deriving a photogenerated carrier density with three significant figures from this order of magnitude estimate is not justified. Measuring or computing a value for the absorption coefficient would provide much firmer support for the numbers derived from these fits.

The authors also assume that the Y₂Ti₂O₅S₂ material is "heavily" n-doped, but provide no explicit justification for this assumption. Previous work (ref 24) contains Mott-Schottky plots that indicate n-type behavior, but they do not quantify the carrier density using these data. Such quantification would also help validate the results of the fits to their TA data.

Finally, although I appreciate the precision of the authors' quantitative analysis of how various parameters impact the performance of Y₂Ti₂O₅S₂ as a photocatalyst for water-splitting, the accuracy of this analysis depends on the values of the input parameters, most of which are derived from the fits of the TA spectra that are themselves based on order of magnitude estimates as discussed above.

In summary, without additional specific evidence to back up the assumptions governing the inputs to their model, it is difficult to ascertain the accuracy of its quantitative outputs. Even with more accurate and precise inputs, it is not clear that this work provides any specific chemical or physical insights into why Y₂Ti₂O₅S₂ is a more effective photocatalyst than other materials, other than that it perhaps has a longer excited state lifetime.

In my opinion, significant revisions to the manuscript to address these points is required.

Reviewer #2 (Remarks to the Author):

Nandal and Shoji et al. present a study of the charge carrier recombination dynamics of the Y₂Ti₂O₅S₂ oxysulfide semiconductor using transient diffuse reflectivity and theoretical modeling to identify the sources of carrier loss that account for the low solar-to-hydrogen conversion efficiency previously reported for this material. Unfortunately, the experimental method chosen is not properly suited to this analysis, as laser-induced heating of solid-state semiconductor materials is well known to have a dramatic impact on optical transient absorption (or reflectivity) signals. See, for example, the following DOIs:

10.1103/PhysRevB.84.195411
10.1039/C6EE02266A
10.1021/acs.jpcc.8b06645
10.1021/acs.jpcc.7b09592

Several of these papers outline possible approaches to account for these effects in transient absorption measurements, but there is no mention of thermal/heating measurements in the main text or SI presented here. In fact, the modeling seems to neglect such contributions to the signal entirely. This is especially problematic given that the power law decay observed at later times is a signature of cooling of such materials.

Thus, while the results presented are quite intriguing and very well may contain the information the authors hoped to present, it is impossible to assess the validity of the analysis without rigorous thermal/heating control measurements and analysis. The authors may wish to pursue other methods for measuring charge carrier dynamics in addition to (or instead of) TDR that are not affected by thermal effects, such as time-resolved THz spectroscopy, time-resolved photoluminescence, or fluorescence upconversion.

Finally, the authors use transient absorption and transient diffuse reflectivity interchangeably throughout the manuscript. This is very confusing and potentially misleading. All data presented are (to my understanding) TDR and not TAS, so they should only be referred to as such.

(* The pages and paragraphs are numbered according to the corrected version of the manuscript.)
(* All changes to the manuscript are highlighted by red color.)

Reviewer 1

This manuscript presents a transient absorption study of $\text{Y}_2\text{Ti}_2\text{O}_5\text{S}_2$, an oxysulfide semiconductor that was recently shown to drive overall water splitting under visible-light irradiation. The authors present some sophisticated modeling of the transient absorption data to describe both the early-time kinetics (attributed to bimolecular recombination of mobile charge carriers) and late-time kinetics, which exhibit power law behavior and are attributed to detrapping of holes from band-tail states. The authors also perform a quantitative analysis of how various material parameters are expected to impact the performance of $\text{Y}_2\text{Ti}_2\text{O}_5\text{S}_2$ in photoinduced hydrogen generation.

Overall, despite the complexity of their model and its ability to fit the TA data, the only insight provided that may explain the improved photocatalytic activity of this specific oxysulfide material compared to other photocatalysts, such as metal oxides, is a longer effective lifetime of the initial photoexcited state of ~6 ns. However, this value and the values obtained for various carrier and trap state densities is based on the assumption that the absorption coefficient is $\sim 10^5 \text{ cm}^{-1}$. Deriving a photogenerated carrier density with three significant figures from this order of magnitude estimate is not justified. Measuring or computing a value for the absorption coefficient would provide much firmer support for the numbers derived from these fits.

Response: We would like to thank the reviewer for suggesting to measure or compute the value of the absorption coefficient of $\text{Y}_2\text{Ti}_2\text{O}_5\text{S}_2$. First, we would like to mention that the characteristic energy E_0 of tail states and effective carrier lifetime $\tau = 1/k_r(n_{eq} + N_t)$ are independent of the absorption coefficient; $k_r(n_{eq} + N_t)$ in Eq. (1) can be determined without knowing the absorption

coefficient. For the determination of k_r (bimolecular recombination rate constant), n_{eq} (equilibrium electron density), and N_t (the total density of states for the shallow traps) from the transient diffuse reflectance spectroscopy (TDRS) analysis, we agree with the reviewer that the absorption coefficient should be measured or computed. As a result, with DFT/HSE06 (HSE06-hybrid functional) calculations, we computed the dielectric function, complex refractive index, reflectivity, and ultimately absorption coefficient of $Y_2Ti_2O_5S_2$. Excellent agreement between the measured reflectivity and computed reflectivity in Figure S6e indicates that the calculated optical properties are reliable. We provide the comparison of absorption coefficient and absorption depth of $Y_2Ti_2O_5S_2$ with other well-studied visible-light absorbing oxysulfide ($La_5Ti_2CuS_5O_7$), metal oxides ($BiVO_4$; $\alpha-Fe_2O_3$) and nitride (Ta_3N_5) photocatalysts. With this, we incorporated the following text on pages 2 and 3 of the main manuscript along with the Figure 1d, Supplementary Figures S5, S6, and S7:

The partial density of states computed using HSE06 hybrid functional (Supplementary Fig. S5) displays an energy band gap of 1.91 eV which is well in agreement with the reported values of 1.9 eV (measured) and 2.19 eV (DFT/HSE06 method).^{24,34} The calculated optical properties such as dielectric function, complex refractive index, reflectivity, and absorption coefficient of $Y_2Ti_2O_5S_2$ are provided in Supplementary Fig. S6. The simulated reflectivity spectra coincide with the measured spectrum which confirms the validity of the computed optical properties. From Fig. 1d or Supplementary Fig. S6f, the absorption coefficient α_{abs} for $Y_2Ti_2O_5S_2$ ranges from 10^3 cm^{-1} to 10^5 cm^{-1} within the ultraviolet-visible solar spectrum from 2.4 eV to 4 eV. In addition, Supplementary Fig. S7 displays that the α_{abs} of $Y_2Ti_2O_5S_2$ is similar to that of other well-studied visible-light absorbing oxysulfide ($La_5Ti_2CuS_5O_7$),³⁵ however, one order less than the metal oxides ($BiVO_4$; $\alpha-Fe_2O_3$)^{36,37} and nitride (Ta_3N_5)^{27,28} photocatalysts. Consequently, the oxysulfides absorb the ultraviolet solar spectrum from 3.1 eV to 4.2 eV within the depth of 50 nm to 600 nm, whereas metal oxides and nitride absorb within 10 nm to 50 nm from the electrolyte interface.

The revised values of β , k_r , n_{eq} , and N_t in the main manuscript text are $1.30 \times 10^{-18} \text{ cm}^3$, $1.57 \times 10^{-10} \text{ cm}^3 \text{ s}^{-1}$, $5.2 \times 10^{17} \text{ cm}^{-3}$, and $5.1 \times 10^{17} \text{ cm}^{-3}$, respectively. With these extracted parameters, we updated the Figures 2, 3, 4, and 5 of the main manuscript along with Supplementary Figures S12 and S13. Revised TDRS analysis leads to minor changes in respective figures. In addition, the horizontal axis of Figure 2b of the main manuscript text is changed from absorbed photon density I_p (photons· cm^{-3}) to incident photon density I_p (photons· cm^{-2}) for the better understanding of readers.

By TDRS analysis up to several ns, k_r and $n_{eq} + N_t$ values can be estimated after the absorption coefficient is determined. The value of each n_{eq} and N_t can be obtained after employing theoretical analysis of TDRS results in μs time range. Analysis shown in Fig. 3b is required to decouple n_{eq} and N_t from the sum $n_{eq} + N_t$ obtained from TDRS analysis up to several ns and theory shown in SI is needed for this analysis. These values and the characteristic energy E_0 of tail states influence the efficiency of photocatalytic activity of $Y_2Ti_2O_5S_2$ as shown in Fig. 5, in addition to the effective carrier lifetime.

Fig. S6. Ab-initio first-principle calculations of DFT/HSE06 (HSE06-hybrid functional) for optical properties of $Y_2Ti_2O_5S_2$ photocatalyst. **a, b,** Simulated real ϵ_r and imaginary ϵ_i components of dielectric function ($\epsilon = \epsilon_r + i\epsilon_i$). **c, d,** Estimated refractive index n_r and extinction coefficient k_i from relations: $2n_r^2 = (\epsilon_r^2 + \epsilon_i^2)^{1/2} + \epsilon_r$; $2k_i^2 = (\epsilon_r^2 + \epsilon_i^2)^{1/2} - \epsilon_r$. **e,** Measured and calculated reflectivity. **f,** Absorption coefficient $\alpha_{abs} (= 4\pi E k_i / hc$, where h and c are Planck constant and speed of light in vacuum, respectively). Normal reflectivity measurements were performed on the crystal face of particle with the size of 10-20 μm . In panel **e**, the reflectivity is calculated from the simulation results of panel **c** and **d** such that $reflectivity = ((n_r - 1)^2 + k_i^2) / ((n_r + 1)^2 + k_i^2)$. The results in panel **e** displays that the measured reflectivity and calculated reflectivity in xx or yy direction are in excellent agreement which suggest accurate estimation of absorption coefficient in panel **f**. In panel **a-f**, xx, yy, and zz correspond to the computed optical parameters along x, y, and z directions of $Y_2Ti_2O_5S_2$ crystal model in Fig. S4a.

The authors also assume that the $Y_2Ti_2O_5S_2$ material is "heavily" n-doped, but provide no explicit justification for this assumption. Previous work (ref 24) contains Mott-Schottky plots that indicate n-type behavior, but they do not quantify the carrier density using these data. Such quantification would also help validate the results of the fits to their TA data.

Response: We would like to thank the reviewer for his/her constructive comment. As suggested by the reviewer, we provide the analysis of extracted Mott-Schottky (MS) plot to determine n-type doping density of $Y_2Ti_2O_5S_2$ in Supplementary Figure S11. To obtain MS plot, the capacitance was normalized to the projected substrate area which was less than the total surface area of the

$\text{Y}_2\text{Ti}_2\text{O}_5\text{S}_2$ particles. Considering this, the extracted doping density from MS analysis provides the upper limit of n-type doping density N_d^{max} . The results indicate that the N_d^{max} is more than 10^{20} cm^{-3} which suggests that the Fermi-energy level E_f is close to the conduction band energy minimum of E_c , as per Fermi-Dirac statistics. MS analysis confirms the validity of our assumption of heavily n-type doped semiconductor. Above discussion is incorporated on page 6 of the main manuscript and Supplementary Figure S11 (shown below).

Figure S11. Mott-Schottky (MS) analysis of $\text{Y}_2\text{Ti}_2\text{O}_5\text{S}_2$ electrode. The reported $\text{Y}_2\text{Ti}_2\text{O}_5\text{S}_2/\text{Ti}$ electrode was prepared by particle-transfer method.⁶ Electrochemical impedance (at frequency 1kHz and AC amplitude 10 mV) was measured in three electrode configurations with Pt and Ag/AgCl as counter and reference electrodes, respectively. An electrolyte solution of 0.1 M Na_2SO_4 was prepared and adjusted by H_2SO_4 (aq.) or NaOH (aq.) for pH of 6.8. The depletion capacitance C at $\text{Y}_2\text{Ti}_2\text{O}_5\text{S}_2$ /electrolyte interface reduces with the increase of applied potential V , following the well-known MS relation given by $C^{-2} = 2(V - V_{FB}) / (qN_d^{\text{max}} \epsilon_r \epsilon_0)$. Here, C , q , ϵ_0 , ϵ_r , N_d^{max} , and V_{FB} are area normalized capacitance, the elementary charge, electrical permittivity of free space, dielectric constant, n-type doping density, and flat band potential, respectively. Using computed $\epsilon_r = 5.12$ at 0 eV (in Fig. S6a), $N_d^{\text{max}} = 1.9 \times 10^{20} \text{ cm}^{-3}$ is calculated from the slope of linear fit (dashed line) to the measured MS plot. The extracted n-type doping density of $\text{Y}_2\text{Ti}_2\text{O}_5\text{S}_2$ is the upper limit as the capacitance was normalized to the projected electrode area which was less than the surface area of $\text{Y}_2\text{Ti}_2\text{O}_5\text{S}_2$ /electrolyte interface.

Finally, although I appreciate the precision of the authors' quantitative analysis of how various parameters impact the performance of $\text{Y}_2\text{Ti}_2\text{O}_5\text{S}_2$ as a photocatalyst for water-splitting, the accuracy of this analysis depends on the values of the input parameters, most of which are derived from the fits of the TA spectra that are themselves based on order of magnitude estimates as discussed above.

Response. We scrutinized the value of the absorption coefficient and the values obtained from TAS analysis such as k_r , n_{eq} , and N_t are revised accordingly. The value of E_0 and the effective carrier lifetime are unaffected. We observe that the re-calculated material parameters are scaled almost linearly from the previously assumed absorption coefficient.

In Figure 5a ,c , and d, we present how various parameters impact the performance of $Y_2Ti_2O_5S_2$ as a photocatalyst for water-splitting by assuming AM 1.5G illumination. The influence of light-intensity is studied in Figure 5b. By considering that the values obtained from the analysis of the TDRS signal are on order of magnitude estimates, all figures in Figure 5 are presented using logarithmic scale for the abscissa and the obtained parameter values are indicated by the dashed lines in Figure 5c and d. Improvement of IQE by reducing n_{eq} depends on the value of N_t . The blue line in Figure 5c is obtained using the estimated N_t value and the other lines are given for the order of magnitude change in N_t value. Similarly, the lines in Figure 5c are drawn using the order of magnitude change in n_{eq} value. Although the analysis of TDRS signal is themselves based on order of magnitude estimates, the performance optimization design strategies, and the potential of $Y_2Ti_2O_5S_2$ as a photocatalyst for water splitting can be studied semi-quantitatively. For example, in Figure 5c, IQE is insensitive to N_t value under the current doping level but increasingly influenced by N_t value by reducing n_{eq} .

In summary, without additional specific evidence to back up the assumptions governing the inputs to their model, it is difficult to ascertain the accuracy of its quantitative outputs. Even with more accurate and precise inputs, it is not clear that this work provides any specific chemical or physical insights into why $Y_2Ti_2O_5S_2$ is a more effective photocatalyst than other materials, other than that it perhaps has a longer excited state lifetime.

Response: The values of E_0 and the effective carrier lifetime are unaffected by the value of the absorption coefficient. These are directly determined without the other input parameters. The absorption coefficient is carefully estimated by the most reliable DFT/HSE06 calculations. The accuracy of its quantitative outputs (k_r , n_{eq} , and N_t) are improved accordingly.

In the previous manuscript, we present optimization design strategies to realize high efficiency for $Y_2Ti_2O_5S_2$ photocatalyst but explanation on why $Y_2Ti_2O_5S_2$ is a more effective photocatalyst than other materials might be inadequate besides a longer excited state lifetime. In the revised manuscript, we added such explanation on the basis of the lists given below:

- 1) $Y_2Ti_2O_5S_2$ photocatalyst has low and direct energy band gap with theoretical efficiency limit of 20.9%.
- 2) The optical properties of $Y_2Ti_2O_5S_2$ are determined by the DFT/HSE06 calculations and the measured reflectance spectra. The absorption coefficient and absorption depth of $Y_2Ti_2O_5S_2$ are similar to those of other visible light absorbing oxysulfide photocatalysts.
- 3) The early-time decay up to few nanoseconds is governed by bimolecular (band-to-band) recombination process with the recombination rate constant $1.57 \times 10^{-10} \text{ cm}^3\text{s}^{-1}$ comparable with the bimolecular recombination rate constant $\sim 0.5 - 1 \times 10^{-10} \text{ cm}^3\text{s}^{-1}$ of direct band-gap semiconductor GaAs, and that $1 \sim 10 \times 10^{-10} \text{ cm}^3\text{s}^{-1}$ of perovskite $CH_3NH_3PbI_3$.
- 4) The n-type doping density (n_{eq}) of the particulate $Y_2Ti_2O_5S_2$ is estimated by our theoretical analysis on the transient diffuse reflectance signals. In the Mott-Schottky (MS) analysis, n_{eq} can be estimated but the capacitance was normalized to the projected Ti substrate area which was less than the total surface area of the $Y_2Ti_2O_5S_2$ /electrolyte interface. Estimation of the value of n_{eq} of the particulate systems could be difficult by the other existing

methods. The situation is in sharp contrast with that of plate electrodes, where the physical properties can be measured by the conventional means such as hall measurement, Mott-Schottky (MS) analysis, and spectroscopic ellipsometry.

- 5) Though the hole trap states are found, they are shallow exponential tail-trap states near valence band ($E_0=0.137\text{ eV}$). Holes can be detrapped by thermal energy and trap-assisted recombination is negligible.

We realized that the points 2) to 4) are not fully addressed in the previous manuscript and related description for these points are added in the revised manuscript on pages 2, 3, 6, and 9. In addition to the above listed points, the previous study reported a stable stoichiometric production of hydrogen and oxygen gases. With these promising characteristics, $\text{Y}_2\text{Ti}_2\text{O}_5\text{S}_2$ photocatalytic material has tremendous potential for commercialization as recently reviewed by three of the authors (Trends in Chemistry 2020, 2, 813) and others (G. Zhang, X. Wang, Angew. Chem. Int. Ed. 2019, 58, 15580). Despite potential for scalability of particulate devices, means to characterize particulate electrodes are limited compared to thin-film-based electrodes; our work presents the crucial and fundamental insights behind the photophysical properties of visible-light absorbing $\text{Y}_2\text{Ti}_2\text{O}_5\text{S}_2$ photocatalyst.

In my opinion, significant revisions to the manuscript to address these points is required.

Response: As suggested by the reviewer, we significantly revised the manuscript by including the computed optical properties including absorption coefficient, measured reflectivity spectrum, re-examined TDRS analysis, extracted physical material parameters, and updated Figures and text.

Reviewer 2

Nandal and Shoji et al. present a study of the charge carrier recombination dynamics of the $\text{Y}_2\text{Ti}_2\text{O}_5\text{S}_2$ oxysulfide semiconductor using transient diffuse reflectivity and theoretical modeling to identify the sources of carrier loss that account for the low solar-to-hydrogen conversion efficiency previously reported for this material. Unfortunately, the experimental method chosen is not properly suited to this analysis, as laser-induced heating of solid-state semiconductor materials is well known to have a dramatic impact on optical transient absorption (or reflectivity) signals. See, for example, the following DOIs:

10.1103/PhysRevB.84.195411

10.1039/C6EE02266A

10.1021/acs.jpcc.8b06645

10.1021/acs.jpcc.7b09592

Several of these papers outline possible approaches to account for these effects in transient absorption measurements, but there is no mention of thermal/heating measurements in the main text or SI presented here. In fact, the modeling seems to neglect such contributions to the signal entirely. This is especially problematic given that the power law decay observed at later times is a signature of cooling of such materials.

Thus, while the results presented are quite intriguing and very well may contain the information the authors hoped to present, it is impossible to assess the validity of the analysis without rigorous thermal/heating control measurements and analysis.

Response: We would like to thank the reviewer for constructive comments on thermal/heating effect to optical transient absorption signals and giving us the related four literatures.

According to the estimation method of pump-induced initial temperature rise in the materials described in these literatures, we tried to estimate the temperature change caused by the pump light pulse in our case as follows.

If all absorbed energy of pump light is converted to heat, these literatures show that the temperature rise can be estimated as

$$\Delta T = Q/\rho C_p,$$

where Q is the absorbed energy of pump light per pulse and volume ($\text{J}\cdot\text{cm}^{-3}$), ρ is the molar density ($\text{mol}\cdot\text{cm}^{-3}$) of $\text{Y}_2\text{Ti}_2\text{O}_5\text{S}_2$, and C_p is the molar heat capacity ($\text{J}\cdot\text{mol}^{-1}\cdot\text{K}^{-1}$) of $\text{Y}_2\text{Ti}_2\text{O}_5\text{S}_2$.

In our study, maximum energy of pump light per pulse is $4.5 \mu\text{J}$ and this corresponds to $0.338 \text{ mJ}\cdot\text{cm}^{-2}$ considering the irradiated area of the pump light. Q is calculated considering the computed absorption coefficient at 3.1 eV ($1.74\times 10^4 \text{ cm}^{-1}$) as

$$Q = 0.338 \times 1.74 \times 10^4 = 5.87 \text{ J}\cdot\text{cm}^{-3}$$

ρ is calculated to be $1.02 \times 10^{-2} \text{ mol}\cdot\text{cm}^{-3}$ from the crystal structure data (Ref.1). C_p of $\text{Y}_2\text{Ti}_2\text{O}_5\text{S}_2$ is experimentally unknown. However, if we assume that the Dulong–Petit law for heat capacity of solids can be applied to $\text{Y}_2\text{Ti}_2\text{O}_5\text{S}_2$, C_p is estimated to be $C_p = 3 \times R \times 11 = 3 \times 8.31 \times 11 = 274.4 \text{ J}\cdot\text{mol}^{-1}\cdot\text{K}^{-1}$, where R is the molar gas constant. (In the paper of 10.1021/acs.jpcc.7b09592, the value of $116 \text{ J}\cdot\text{mol}^{-1}\cdot\text{K}^{-1}$ ($110 \text{ J}\cdot\text{mol}^{-1}\cdot\text{K}^{-1}$) is used for C_p of LaFeO_3 (LaMnO_3). These values are very close to $3 \times 8.31 \times 5 = 124.7 \text{ J}\cdot\text{mol}^{-1}\cdot\text{K}^{-1}$, which is expected from the Dulong–Petit law.) Substituting these values into the above equation of ΔT , we obtain $\Delta T = 2.1 \text{ K}$ for maximum energy of pump light per pulse ($4.5 \mu\text{J}/\text{pulse}$). For the measurement of TDR spectra shown in Fig. 1d, the energy of pump light per pulse is $3.0 \mu\text{J}/\text{pulse}$. In this case, ΔT is estimated to be 1.4 K . Thus, the maximum temperature rise is found to be not significant ($\leq 2.1 \text{ K}$) in the case of $\text{Y}_2\text{Ti}_2\text{O}_5\text{S}_2$.

As the reviewer and these literatures point out, in transient absorption and reflectivity measurements, transient heating of the sample induced by the pump laser pulse can cause thermal components in the measured spectra and a proper assessment and isolation of its components are essential for accurate interpretation of photoinduced (nonthermal) electronic responses. Particularly, as highlighted by these literatures, temperature rise causes an energy shift of band gap and broadening of absorption and reflectivity band of sample, and thermal-induced signal often appears near the band-gap edge and above the band gap and is not observed in energy range far below band-gap energy. In fact, the papers of 10.1103/PhysRevB.84.195411 (pentacene), 10.1021/acs.jpcc.8b06645 (BiVO_4), and 10.1021/acs.jpcc.7b09592 (LaFeO_3) clearly show that transient absorption and reflectivity signals in energy range far below band-gap energies ($< 1.7 \text{ eV}$ for pentacene, $< 2.2 \text{ eV}$ for BiVO_4 and $< 2.0 \text{ eV}$ for LaFeO_3) are not thermal-induced signals but can be assigned to photoinduced (nonthermal) electronic signals from comparison between temperature-induced differential spectra and the observed pump-induced transient spectra. This means that transient absorption and reflectivity measurements can capture the accurate photoinduced electronic responses when we select proper probe energy range. Therefore, as in

these cases, transient signals in energy range far below band-gap energy (1.9 eV for $\text{Y}_2\text{Ti}_2\text{O}_5\text{S}_2$) can be judged to be due to the photoinduced electronic signals in the case of $\text{Y}_2\text{Ti}_2\text{O}_5\text{S}_2$.

To further confirm this notion experimentally, we measured steady-state diffuse reflectance spectra of $\text{Y}_2\text{Ti}_2\text{O}_5\text{S}_2$ over a temperature range from 296.5 K to 316.5 K using a Fourier transform infrared spectrometer (FT/IR-6100, JASCO) with an optical microscope (IRT-5000, JASCO) and a cryostat (Microstat, Oxford Instruments). Figure S9 shown below is the differential diffuse reflectance spectra defined by $1-R_T/R_{296.5\text{K}}(\%)$ in the energy range from 0.125 eV to 0.625 eV. The probe photon energy of 0.24 eV at which we measured detailed kinetics up to 1 μs (Fig. 1f in the main text) is located within this range. As can be seen from the figure, no thermal-induced signal is observed within the range of uncertainty ($\pm 1\%$) even when we increase the sample temperature by 20 K from 296.5 K, which is much larger than the above-estimated maximum temperature rise (2.1 K) by pump laser pulse. This result clearly indicates that thermal effect did not reproduce the observed transient spectra shown in Fig. 1e in the main text, and transient spectra from 0.125 eV to 0.625 eV is surely due to pure photoinduced electronic responses.

Furthermore, as mentioned in the main text (page 3), the temporal profiles of absorption signal are found to be almost identical across the entire probe photon energy spectrum (0.15 eV \sim 2.87 eV, Fig. 1e in the main text), indicating that thermal contribution to the observed transient spectra is quite minor. In addition, we present the TDR kinetics (up to 1 μs) probed at 1.48 eV for various pump fluences in Fig. S10 shown below. As can be seen, the kinetics feature at 1.48 eV is almost the same as that at 0.24 eV (Fig. 1f in the main text), suggesting that kinetics feature observed at 0.24 eV is universal, independent of probe photon energy.

From above discussion, therefore, it can be concluded that the TDR signal in $\text{Y}_2\text{Ti}_2\text{O}_5\text{S}_2$ probed at 0.24 eV reflects only photoinduced electronic processes and we think that interpretation and analysis based on this result is valid and reliable.

Fig. S9 Temperature-induced differential diffuse reflectance spectra of $\text{Y}_2\text{Ti}_2\text{O}_5\text{S}_2$ in mid-IR range.

Fig. S10 TDR kinetics at probe energy of 1.48 eV in $Y_2Ti_2O_5S_2$.

As suggested by the reviewer, we give the description on the effect of pump-induced heating on transient signal in the main text (page 4) and SI (page 7-9) on the basis of above discussion.

The authors may wish to pursue other methods for measuring charge carrier dynamics in addition to (or instead of) TDR that are not affected by thermal effects, such as time-resolved THz spectroscopy, time-resolved photoluminescence, or fluorescence upconversion.

Response: We would like to thank the reviewer for proposing other methods for measuring charge carrier dynamics such as time-resolved THz spectroscopy, time-resolved photoluminescence, or fluorescence up-conversion. However, unfortunately, we do not have these instruments and are not available to these methods. Therefore, we give up trying these methods at present. From the proper assessment of pump-induced thermal effect to transient signal as mentioned above, it can be concluded that the TDR signal in $Y_2Ti_2O_5S_2$ probed at 0.24 eV reflects only photoinduced electronic processes and we think that interpretation and analysis based on this result is valid and reliable.

Finally, the authors use transient absorption and transient diffuse reflectivity interchangeably throughout the manuscript. This is very confusing and potentially misleading. All data presented are (to my understanding) TDR and not TAS, so they should only be referred to as such.

Response: As the reviewer point out, all data presented in the paper is TDRS (or TDR) and not TAS (or TA). Following the reviewer's suggestion, we use TDRS (or TDR) throughout the main text and SI to avoid the potential misleading of readers.

Ref.1 Denis, S. G & Clarke, S. J. Two alternative products from the intercalation of alkali metals into cation-defective Ruddlesden–Popper oxysulfides. *Chem. Commun.* 2356-2357 (2001).

REVIEWERS' COMMENTS

Reviewer #2 (Remarks to the Author):

The authors have sufficiently addressed my primary concern, namely the photoinduced heating of the sample and its effects on the TDR spectra reported. It would have been preferable to see the temperature-dependent DR spectra at visible wavelengths, but given that the kinetic analysis is all based upon the transients measured at 0.24 eV, this is not strictly necessary. (Note: the time traces shown in Fig. 1f and 3a are explicitly stated to be at a probe energy of 0.24 eV, but this is not stated for Fig. 2a. I assume this is the case, but it should be stated in the figure caption and/or main text). Given this, and the fact that the manuscript presents a very thorough characterization of a promising new photocatalyst, I recommend accepting it for publication in Nature Communications following minor revisions as described below.

1) The authors calculate a direct band gap of 0.84 eV (which they state is underestimated) using a PBE hybrid functional (I believe?) in Fig. S4, and then they perform another DFT calculation with the HSE06 hybrid functional and get a more reasonable value of 1.91 eV. Both calculations clearly show that the "conduction band (CB) minimum and valence band (VB) maximum are primarily contributed from Ti-3d and S-3p orbitals, respectively." From what I can tell, however, the first calculation is never mentioned again, other than to show that the material has a direct band gap. Would it not be better to repeat the band structure calculation using the same method as in the second calculation (which gave a much more accurate band gap) to obtain better results overall? Additionally, the fact that there is such a stark difference suggests that either the first calculation is unreliable or both calculations are. Is there a reason to expect that the second calculation is reliable, beyond just the agreement with the experimental band gap energy and reflectivity spectrum?

2) Along similar lines, the authors state that the 1.91 eV band gap they calculate is in good agreement with the value reported in ref. 34 (2.19 eV). This calculation also used the HSE06 hybrid functional, so the authors should explain what the origin of this disparity is.

3) The method for estimating the photoinduced temperature jump is averaged over a thickness of 1 cm, while 33% of the incident energy is absorbed within the first ~100 nm of material (which is also the sample thickness reported in the SI). Thus, it makes more sense to calculate the energy density for a 100 nm thick absorber, which by my calculations is 11.2 J/cm³. This is only about twice the value the authors find, so it doesn't have a major impact on their discussion of thermal effects, but this should be corrected.

Reviewer 2

The authors have sufficiently addressed my primary concern, namely the photoinduced heating of the sample and its effects on the TDR spectra reported. It would have been preferable to see the temperature-dependent DR spectra at visible wavelengths, but given that the kinetic analysis is all based upon the transients measured at 0.24 eV, this is not strictly necessary. (Note: the time traces shown in Fig. 1f and 3a are explicitly stated to be at a probe energy of 0.24 eV, but this is not stated for Fig. 2a. I assume this is the case, but it should be stated in the figure caption and/or main text). Given this, and the fact that the manuscript presents a very thorough characterization of a promising new photocatalyst, I recommend accepting it for publication in Nature Communications following minor revisions as described below.

Response: We would like to thank the reviewer for his positive evaluation and careful reading of the manuscript. In the caption of Fig. 2a, “**at probe energy of 0.24 eV.**” is added.

1) *The authors calculate a direct band gap of 0.84 eV (which they state is underestimated) using a PBE hybrid functional (I believe?) in Fig. S4, and then they perform another DFT calculation with the HSE06 hybrid functional and get a more reasonable value of 1.91 eV. Both calculations clearly show that the “conduction band (CB) minimum and valence band (VB) maximum are primarily contributed from Ti-3d and S-3p orbitals, respectively.” From what I can tell, however, the first calculation is never mentioned again, other than to show that the material has a direct band gap. Would it not be better to repeat the band structure calculation using the same method as in the second calculation (which gave a much more accurate band gap) to obtain better results overall? Additionally, the fact that there is such a stark difference suggests that either the first calculation is unreliable or both calculations are. Is there a reason to expect that the second calculation is reliable, beyond just the agreement with the experimental band gap energy and reflectivity spectrum?*

Response: The band structure was already shown in Ref. 34 using DFT/HSE06 method and is very similar to what we presented in this manuscript except the value of band-gap energy. It is known that the band-gap energy is underestimated by PBE due to the functional derivative discontinuity of the exchange-correlation potential energy. [L. J. Sham and M. Schlüter, Phys. Rev. Lett. **51**, 1888 (1983)] Therefore, in the revised manuscript, we added the following text on page 2 of the main manuscript:

"Band structure calculation in Supplementary Fig. S4b reveals that $\text{Y}_2\text{Ti}_2\text{O}_5\text{S}_2$ is a direct band gap semiconductor as pointed out previously.³⁴"

2) *Along similar lines, the authors state that the 1.91 eV band gap they calculate is in good agreement with the value reported in ref. 34 (2.19 eV). This calculation also used the HSE06*

hybrid functional, so the authors should explain what the origin of this disparity is.

Response: The origin of above-mentioned disparity is originated from the different basis set employed in our work and that reported in ref. 34. In our work, we utilized plane-wave basis set, whereas the authors of ref. 34 employed atom-centred Gaussian basis sets for calculations. To highlight the origin of discrepancy, we provide the following manuscript on page 3 of the main manuscript:

"The slight discrepancy in the values of the band gap might be attributed to the difference in the basis sets. "

3) *The method for estimating the photoinduced temperature jump is averaged over a thickness of 1 cm, while 33% of the incident energy is absorbed within the first ~100 nm of material (which is also the sample thickness reported in the SI). Thus, it makes more sense to calculate the energy density for a 100 nm thick absorber, which by my calculations is 11.2 J/cm³. This is only about twice the value the authors find, so it doesn't have a major impact on their discussion of thermal effects, but this should be corrected.*

Response: We would like to thank the reviewer for his comment on estimating the photoinduced temperature jump. We have re-checked the estimation procedure as follows.

According to the Beer-Lambert law, at give depth L, the absorbed ratio of incident pump energy is expressed by $1 - \exp(-\alpha_{\text{pump}}L)$, where α_{pump} is absorption coefficient at pump wavelength (400 nm). In our case, $\alpha_{\text{pump}} = 17400 \text{ cm}^{-1}$. Therefore, within the first L= 100 nm depth of material, about 16 % of the incident pump energy is absorbed because $1 - \exp(-\alpha_{\text{pump}}L) = 1 - \exp(-0.00174 * 100) = 1 - \exp(-0.174) = 0.1597$. We guess that the reviewer may misunderstand the Beer-Lambert law and used the following equation, that is, $1 - 10^{-(\alpha_{\text{pump}}L)} = 1 - 10^{-0.174} = 0.3301$.

Using the value of depth L=100 nm and absorbed ratio of incident pump energy of 16 %, we can calculate the energy density for a 100 nm thick absorber using the same method mentioned by the reviewer; the value is 5.40 J/cm³ which is almost the same as the value (5.87 J/cm³) shown in SI.

From above discussion, therefore, we judge the estimated value of photoinduced temperature jump shown in SI is correct and do not revised the description for this point.